# Molecular Topology for the Discovery of New Broad-Spectrum Antibacterial Drugs

**DOI:** 10.3390/biom10091343

**Published:** 2020-09-19

**Authors:** Jose I. Bueso-Bordils, Pedro A. Alemán-López, Beatriz Suay-García, Rafael Martín-Algarra, Maria J. Duart, Antonio Falcó, Gerardo M. Antón-Fos

**Affiliations:** 1Departamento de Farmacia, Universidad Cardenal Herrera-CEU, CEU Universities C/Ramón y Cajal s/n, 46115 Alfara del Patriarca (Valencia), Spain; paleman@uchceu.es (P.A.A.-L.); rmartin@uchceu.es (R.M.-A.); mduart@uchceu.es (M.J.D.); ganton@uchceu.es (G.M.A.-F.); 2ESI International Chair@CEU-UCH, Departamento de Matemáticas, Física y Ciencias Tecnológicas, Universidad Cardenal Herrera-CEU, CEU Universities San Bartolomé 55, 46115 Alfara del Patriarca (Valencia), Spain; beatriz.suay@uchceu.es (B.S.-G.); afalco@uchceu.es (A.F.)

**Keywords:** antibacterial, antibiotics, computational chemistry, linear discriminant analysis, molecular topology, molecular connectivity, topological indices, quinolones, QSAR

## Abstract

In this study, molecular topology was used to develop several discriminant equations capable of classifying compounds according to their antibacterial activity. Topological indices were used as structural descriptors and their relation to antibacterial activity was determined by applying linear discriminant analysis (LDA) on a group of quinolones and quinolone-like compounds. Four equations were constructed, named DF1, DF2, DF3, and DF4, all with good statistical parameters such as Fisher–Snedecor’s F (over 25 in all cases), Wilk’s lambda (below 0.36 in all cases) and percentage of correct classification (over 80% in all cases), which allows a reliable extrapolation prediction of antibacterial activity in any organic compound. From the four discriminant functions, it can be extracted that the presence of sp^3^ carbons, ramifications, and secondary amine groups in a molecule enhance antibacterial activity, whereas the presence of 5-member rings, sp^2^ carbons, and sp^2^ oxygens hinder it. The results obtained clearly reveal the high efficiency of combining molecular topology with LDA for the prediction of antibacterial activity.

## 1. Introduction

The synthesis of artificial antibacterial agents and the discovery and improvement of antibiotics have brought about a true pharmacological revolution in the treatment of infectious diseases in this century. However, the extreme versatility and adaptability of microorganisms has prevented a decrease in the prevalence of infectious diseases, since many bacteria have been developing mechanisms that protect them against many drugs [1].

Currently, there is a growing concern about decreased efficacy of antimicrobial agents and increased prevalence of new and old bacterial pathogens. Increases in the rate of antibiotic resistance are resulting in higher mortality rates, and higher healthcare costs [2]. In fact, according to the World Health Organization (WHO), bacterial resistances will be the leading cause of death in 2050 and the biggest challenge in the field of Biomedicine in the 21st century [3]. For all these reasons, it is necessary to know the sensitivity of the main microorganisms and to be continuously alert to the appearance of resistant strains that could lead to treatment failure [2].

Given the problem involved in antibacterial therapy, the emergence of resistance to treatments with classic antibacterial compounds, it is necessary to expand the therapeutic arsenal of any group of antibacterial agents. One effective and low-cost way of tackling this problem is by using molecular connectivity or molecular topology (MT), developed by Kier and Hall in the mid-1970s, a Quantitative Structure-Activity Relationships (QSAR) derived method capable of predicting molecular properties in new compounds, without the need to synthesize them [4]. By combining it with techniques of pattern recognition such as linear discriminant analysis (LDA) [5], neural networks [6], multilinear regression [7] or principal component analysis [8] and appropriately selecting the molecular descriptors to use, we can build mathematical-topological equations able to identify almost any molecular property, becoming a powerful tool for the search and design of new compounds with pharmacological activities. 

In this context, this study aims to obtain mathematical-topological equations capable of predicting antibacterial activity. By combining MT and LDA, the topological indices (TI) can be used to classify a compound as antibacterial or non-antibacterial. To do this, we simply select a group of compounds with antibacterial activity and another one lacking it.

To do this, antibacterial and non-antibacterial quinolones have been selected, since it is a known and extensive group that will allow us to collect numerous data [9], leading to greater precision of the predictive equations [10]. Moreover, the interest of the therapeutic target group is reflected in the large number of publications issued since 2010, more than 21,000 articles according to PubMed and 8000 according to the Web of Science (WoS). The discriminant equations obtained can be applied both in new quinolones not used in the study as in molecules that have no structural relationship, since it will select those that have a similar mathematical-topological relationship. The effectiveness of the topological method has also been proven, having been successfully applied in different therapeutic groups such as oral hypoglycemic [11], anti-inflammatory [12], antimalarial [13], antihistaminic [14], antidementia [15] and antibacterial agents [16].

## 2. Materials and Methods

### 2.1. Compound Selection

With the purpose of collecting the maximum information on experimental values, we held a comprehensive and exhaustive bibliographical review. We collected information on quinolones using the ISI Web of Science, Medline and SciFinder (Caplus) search engines.

We finally collected in vitro activity data of 99 quinolones and structurally related compounds against various bacteria, which were classified into two groups: one with 50 antibacterial quinolones and one with 49 non-antibacterial quinolones or closely related structures (the structure of all compounds as well as bibliographic references about their activity can be found in Appendix A). To consider a compound as antibacterial, it should be active (minimum inhibitory concentration (MIC) < 1 mg/mL) against numerous bacteria, while those compounds considered non-antibacterial (MIC >16 mg/mL) should be inactive against at least three Gram-positive and three Gram-negative bacteria. Those with intermediate activities were not included in the study. Regarding stereoisomers, if any of them was active, it was included in the active group. If all individual stereoisomers or the mixture of them in any ratio was inactive, they were included as a single graph in the inactive group.

### 2.2. Topological Descriptors

Each compound was characterized by a set of 144 non-redundant, significant (indices with value 0 for every compound and with identical values for all quinolones were removed) descriptors specific to each molecule. These descriptors do not contain 3D parameters that distinguish between enantiomers. They were computed from the adjacency topological matrix obtained from the hydrogen-depleted chemical pseudographs, previously drawn with the ChemBioDraw Ultra 12.0 drawing program (ChemBioOffice Ultra 2010, CambridgeSoft: 100 CambridgePark Drive, Cambridge, MA, USA 02140: 2010) by using MOLCONN-Z (MOLCONN-Z software, Eastern Nazarene College: Quincy (MA, USA), 1995) and DESMOL13 (DESMOL13 software, Unidad de Investigación de Diseño de Fármacos y Conectividad Molecular, Facultad de Farmacia, Universitat de València: Valencia (Spain), 2000) programs.

Chemical pseudographs are two-dimensional objects and, therefore, so are their molecular descriptors, although they can represent a high content of information about three-dimensional structures [17]. The molecular descriptors used are described in Appendix A along with their definitions and references. 

### 2.3. Linear Discriminant Analysis (LDA)

Stepwise linear discriminant analysis, LDA, is a pattern recognition method providing a classification model based on the combination of variables that best predicts the category or group to which a given compound belongs. The variables used to compute the linear classification functions were chosen in a stepwise manner, based on the Fisher–Snedecor parameter F, which relates the variance explained by the equation with the residual variance. At each step, the variable with the greater value of F, thus, the variable that makes the larger contribution to the separation of the groups, was entered in the discriminant function. Conversely, selected variables with a small value of F, thus, variables which lowered the statistical significance of the classification function, were removed.

The discriminant ability was assessed by the percentage of correct classifications attained for each set. The classification criterion is the minimal Mahalanobis distance (distance of each case to the mean of all the cases in a category). The quality of the discriminant function was evaluated through Wilk’s U-statistical parameter, λ, which was obtained by a multivariate analysis of variance that tests the equality of group means for the variable in the discriminant model. LDA was then applied to the database, except for the molecules reserved as the test group, to obtain a predictive mathematical model linking structural descriptors and activity. The independent variables in this study were the topological descriptors, and the discriminant property was antimicrobial activity. The software used for the LDA study was the BioMeDical Program (BMDP) 7 M module (BMDP Statistical software Inc., University of California: Berkeley, CA, USA, 1990), which randomly chooses the compounds reserved for the test set.

The equations were finally validated internally by the Jack-Knife (JK) method and externally by the test groups.

### 2.4. Pharmacological Distribution Diagrams

After selecting the discriminant function, the corresponding pharmacological distribution diagrams (PDD) were built up. These plots are useful to determine the intervals of the discriminant function in which the expectancy, E, to find active compounds is maximum. PDDs are histogram-like plots of connectivity functions in which the expectancies appear on the ordinate axis. For an arbitrary interval of values of a given function, we can define the expectancy of activity as Ea = a/(i + 1), where “a” is the number of active compounds in the interval divided by the total number of active compounds, and “i” is the number of inactive compounds. The expectancy of inactivity is defined in a symmetrical way, as Ei = i/(a + 1). This representation provides good visualization of the regions of minimum overlap and selects regions in which the probability of finding improved compounds is maximum [18].

PDDs allowed us to carry out the assignment of thresholds useful to discriminate active from inactive compounds with the highest probability of success. 

## 3. Results

To obtain the discriminant functions we used the data from a previous study [19], in which the statistical program BMDP randomly formed two training groups with 38 active and 38 inactive compounds and two test groups with 12 active and 11 inactive compounds. This test groups allowed evaluating the quality of the selected discriminant functions.

The discriminant functions formed in this study along with their statistical parameters are shown in Equations (1) to (4), while their corresponding classification matrices are shown in Table 1, Table 2, Table 3 and Table 4.

DF1 = 6.51816 − 2.13276*G*_1_ − 4.21995*G*_4_ + 246.05433*J*_5_ + 6.10518^0^*D* − 9.24587^3^*C_c_* N = 76 λ = 0.3545599 F = 25.486 (1)
DF2 = −2.70225 + 14.37228^3^*χ_ch_* − 3.22125*S_=C<_* + 0.99321*S_>C<_* − 0.19709*_S=O_*N = 76 λ = 0.3437429 F = 33.887 (2)
DF3 = −10.33912 + 11.61779^3^*χ^V^_c_* − 44.2419^5^*χ^V^_ch_* − 5.66785*S_=C<_* + 2.35012*S_>C<_* + 1.1054*S_−NH−_* − 0.46032*S_=O_* N = 76 λ = 0.2474651 F = 34.971(3)
DF4 = 22.9069 − 35.07866^5^*χ^V^_ch_* − 6.27951*S_=C<_* + 1.18678*S_−NH−_* − 0.32317*S_=O_* − 11.66095^3^*C_c_*N = 76 λ = 0.254795 F = 40.946(4)

The classification criterion was determined by the value of the discriminant functions: if the value of the equation for a given compound was equal or bigger than 0, such compound was classified as active, whereas if the value of the equation for a compound was smaller than 0, such compound was classified as inactive.

We plotted the corresponding PDDs for every function to visualize the values of the function in which the probability of classifying a compound as active or inactive is maximum. In other words, to find areas where the overlap between the two groups of compounds is minimal. The PDDs obtained for DF1-4 along with the highest activity range for each function are shown in Figure 1, Figure 2, Figure 3 and Figure 4. Compounds with values below the range were considered inactive while compounds with values over the range were considered unclassified. Thus, the value ranges derived from these PDDs establish the applicability domain for each of the discriminant functions.

When PDDs are used, the accuracy for active compounds decreases whereas that for inactive ones increases, so the probability of a false active compound being selected after applying the PDD filter decreases. The average percentage for the four equations of correctly classified inactive compounds is 94.1% for the training group and 100% for the test group; and the average percentage of accurately classified active compounds is 88.8% for the training group and 87.5% for the test group. Table 5 and Table 6 summarize the classification of the results obtained for all functions selected for both training groups, active group and inactive group, respectively, and Table 7 summarizes the results for the test groups. As can be inferred from the tables, the training and test groups exhibit an average overall accuracy of 91.9%. 

## 4. Discussion

All equations have a low value of λ, indicating that there is a low linear dependence between independent variables. Furthermore, the high value of F in the equations indicate that the selected independent variables contribute largely to the separation of the active and inactive groups. Moreover, all equations correctly classify each compound with its corresponding group with high success rates.

Three functions (DF1-3) were obtained using combinations of different types of indices: DF1 used charge indices [20] and connectivity’s differences and ratios indices, while DF2 and DF3 used electrotopological-state (*S_i_*) [21] and connectivity indices. DF4 was obtained using all 144 TI.

DF1 involves three charge indices (*G*_1_, *G*_4_ and *J*_5_) and two connectivity indices formed by difference and ratios between non-valence and valence connectivity indices of zero order (^0^*D*) and third order cluster type (^3^*C_c_*). The DF1 value is influenced very positively by the topological charge present in 5th order subgraphs (*J*_5_), but negatively by the topological charge found in the 1st and 4th order subgraphs (*G*_1_ and *G*_1_). Differences (^0^*D*) and ratios (^3^*C_c_*) of non-valence and valence indices affect the value of DF1 oppositely. The first type of indices describe the distribution of the global charge in the molecule through the evaluation of charge transfer between pairs of atoms, while the second type of indices take into account the charge densities from each chemical graph, representing a measure of the polarizability of the molecule.

DF2 involves one connectivity index (^3^*χ_ch_*) and three electrotopological indices (*S_=C<_*, *S_>C<_* and *S_=O_*). In the case of DF2, the presence of sp^3^ carbons (*S_>C<_*) increase its value while the presence of sp^2^ carbons (*S_=C<_*), and sp^2^ oxygens (*S_=O_*) decreases it. The equation also shows a clear dependence of the activity relative to the connectivity chain type third order index (^3^*χ_ch_*), which implies that the presence of a cyclopropyl group greatly enhances the antibacterial activity.

DF3 involves two valence connectivity indices (^3^*χ^V^_c_* and ^5^*χ^V^_ch_*) and four electrotopological indices (*S_=C<_*, *S_>C<_*, *S_-NH-_* and *S_=O_*). In this case, the value of the equation is positively influenced by the valence connectivity cluster type third order index (^3^*χ^V^_c_*), meaning that the presence of ramifications (especially if they are attached to hydrogen donors) favors the antibacterial activity. On the other hand, the value of the equation is very negatively influenced by the valence connectivity chain type fifth order index (^5^*χ^V^_ch_*), meaning that the presence of 5-member rings (especially if such rings include double bonds and/or hydrogen donors). Regarding the electrotopological indices, the presence of sp^3^ carbons (*S_>C<_*) and secondary amine groups (*S_-NH-_*) increase the value of the equation while the presence of sp^2^ carbons (*S_=C<_*) and sp^2^ oxygens (*S_=O_*) decrease it.

DF4 involves two connectivity indices (^5^*χ^V^_ch_*and ^3^*C_c_*) and three electrotopological indices (*S_=C<_*, *S_-NH-_* and *S_=O_*). The connectivity indices are the valence connectivity chain type fifth order index (^5^*χ^V^_ch_*) and the ratio between non-valence and valence connectivity indices of third order cluster type (^3^*C_c_*). The only index that has a positive influence on the value of the equation is *S_-NH-_*, meaning that the presence of secondary amine groups favors the antibacterial activity, while the rest are detrimental for such activity.

MT transforms the structure into numbers by applying more or less complex mathematical functions [4], so it is generally difficult to draw structural conclusions from mathematical equations, but from the four discriminant functions it can be extracted that the presence of sp^3^ carbons, ramifications, and secondary amine groups in a molecule enhance antibacterial activity, whereas the presence of 5-member rings, sp^2^ carbons, and sp^2^ oxygens hinder it.

Furthermore, these discriminant functions can be used for the development of new antibacterial drugs in different ways. First, QSAR models are traditionally used to optimize the activity of already existing molecules. Along these lines, Wang et al. [22] developed a QSAR model based on cinnamaldehyde-amino acid Schiff base compounds, a family of newly discovered compounds that exhibit good broad-spectrum antibacterial activity, which was then used to design and synthesize three new compounds with comparable antibacterial activity to that of ciprofloxacin. Similarly, De Bruijn et al. [23] used the information derived from a QSAR model to develop novel lead compounds with a 1,4-benzoxazin-3-one scaffold which were reported to be up to 5 times more active than any of the analogue compounds used to build the model. In the same way that these authors used the topological information provided by their models to increase the activity of a family of compounds, the DFs built in this work also provide information which could be useful to optimize the activity of antibacterial quinolones.

The second main approach by which these DFs could aid in the discovery of new antibacterial compounds is drug repurposing. There is extensive literature in which using QSAR models to reposition approved drugs is posed as a key tool in the development of new drug-disease associations [24,25]. More specifically, this approach has been already successfully used to identify antibacterial activity in several drugs approved for different purposes (Table 8) [26,27,28,29,30,31].

## 5. Conclusions

Currently, the development of antibiotic resistance in microorganisms is one of the most important problems that have appeared in recent years in the treatment of infectious diseases. MT has proven to be a useful methodology for identifying new compounds with antibacterial activity. By combining it with LDA, four statistically significant discriminant equations with a very good classification success rate have been obtained. The stability of the selected functions is supported by external and internal validation studies, as well as by all the statistical parameters that have been used in their selection, which can be classified as very satisfactory in all cases. These equations, used both as predictive models or as filters, can be extremely useful to find new molecules with antibacterial activity and even for use in drug repositioning, where we find thousands of candidate compounds to be tested for new pharmacological activities. 

We can conclude that the discriminant functions obtained confirm MT as a powerful, cost-effective, and useful tool in the prediction of antibacterial activity. These equations, used either as predictive models or as filters such as Lipinski’s, can be extremely useful to find new molecules with antibacterial activity and even for use in drug repositioning, where we find thousands of candidate compounds to be tested for new pharmacological activities. With this information, it is simpler to understand how certain non-obvious molecular properties described by TI may affect the antibacterial activity of a compound. One additional advantage of MT is that it allows virtual screening of large databases in a short time for the search of possible antibacterial compounds. Moreover, this methodology also allows the prediction of pharmacokinetic and toxicological properties to identify safer and more efficient drugs.

## Figures and Tables

**Figure 1 biomolecules-10-01343-f001:**
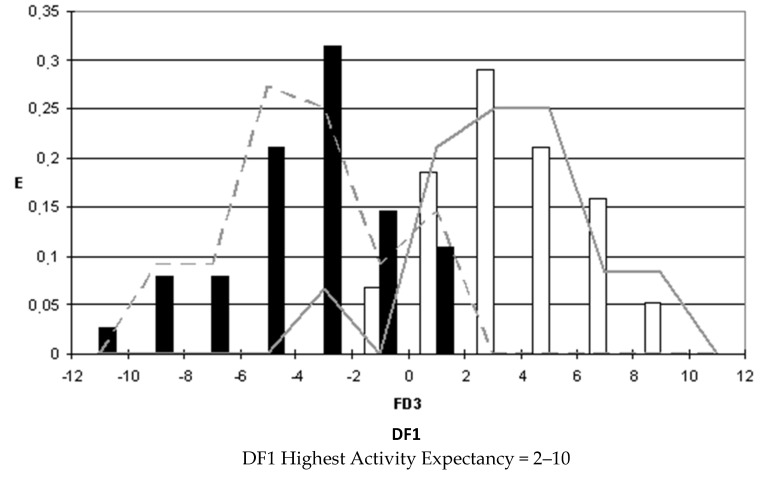
DF1 pharmacological distribution diagram. Black bars: Training Inactives. White bars: Training Actives. Dashed line: Test Inactives. Straight line: Test Actives.

**Figure 2 biomolecules-10-01343-f002:**
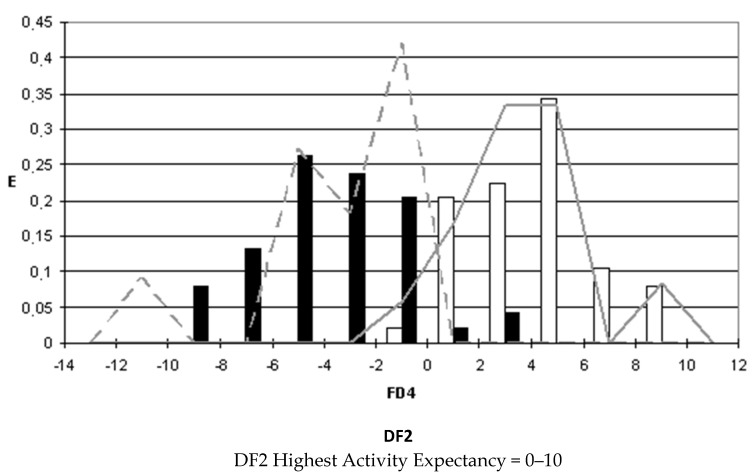
DF2 pharmacological distribution diagram. Black bars: Training Inactives. White bars: Training Actives. Dashed line: Test Inactives. Straight line: Test Actives.

**Figure 3 biomolecules-10-01343-f003:**
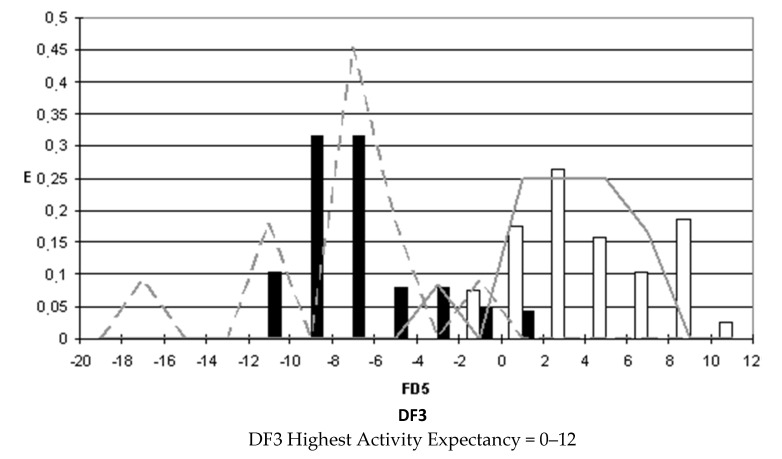
DF3 pharmacological distribution diagram. Black bars: Training Inactives. White bars: Training Actives. Dashed line: Test Inactives. Straight line: Test Actives.

**Figure 4 biomolecules-10-01343-f004:**
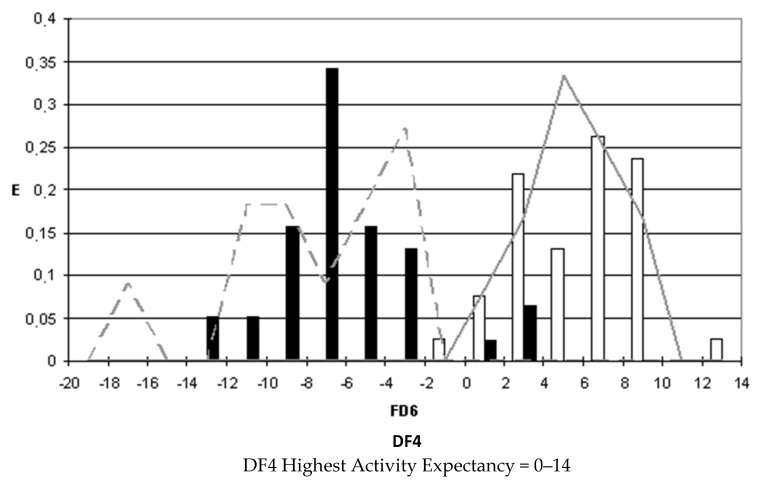
DF4 pharmacological distribution diagram. Black bars: Training Inactives. White bars: Training Actives. Dashed line: Test Inactives. Straight line: Test Actives.

**Table 1 biomolecules-10-01343-t001:** Classification matrix for DF1.

Group	Active	Inactive	% Success
Training active	35	3	92.1
Training inactive	5	33	86.8
Test active	11	1	91.7
Test inactive	2	9	81.8
JK Training active	35	3	92.1
JK Training inactive	5	33	86.8

**Table 2 biomolecules-10-01343-t002:** Classification matrix for DF2.

Group	Active	Inactive	% Success
Training active	37	1	97.4
Training inactive	3	35	92.1
Test active	11	1	91.7
Test inactive	0	11	100
JK Training active	35	3	92.1
JK Training inactive	6	32	84.2

**Table 3 biomolecules-10-01343-t003:** Classification matrix for DF3.

	Active	Inactive	% Success
Training active	36	2	94.7
Training inactive	3	35	92.1
Test active	11	1	91.7
Test inactive	0	11	100
JK Training active	35	3	92.1
JK Training inactive	4	34	89.5

**Table 4 biomolecules-10-01343-t004:** Classification matrix for DF4.

Group	Active	Inactive	% Success
Training active	37	1	97.4
Training inactive	4	34	89.5
Test active	12	0	100
Test inactive	0	11	100
JK Training active	36	2	94.7
JK Training inactive	4	34	89.5

**Table 5 biomolecules-10-01343-t005:** Results obtained after combining LDA and PDD for DF1-4. Training group: actives.

Compound	DF1 ^1^	Clas_DF1_ ^2^	DF2 ^1^	Clas_DF2_ ^2^	DF3 ^1^	Clas_DF3_ ^2^	DF4 ^1^	Clas_DF4_ ^2^
ABT-492	4.950	+	3.886	+	8.906	+	6.043	+
Amifloxacin	1.018	−	0.473	+	3.797	+	6.610	+
BAYy3118	1.899	−	4.249	+	4.345	+	8.537	+
BMS-340278	−1.084	−	2.272	+	0.571	+	3.836	+
BMS-340280	4.423	+	6.162	+	7.714	+	6.312	+
CFC-222	4.369	+	4.355	+	8.364	+	8.289	+
Clinafloxacin	3.267	+	4.516	+	−1.173	−	2.194	+
CS-940	6.137	+	6.079	+	8.585	+	7.711	+
Difloxacin	0.888	−	1.030	+	3.058	+	3.148	+
DQ-113	7.031	+	8.656	+	8.852	+	8.107	+
DW-116	1.281	−	1.345	+	3.365	+	3.254	+
DW286	5.321	+	1.942	+	1.597	+	3.437	+
DX-619	3.772	+	6.674	+	2.660	+	3.626	+
E-4441	8.088	+	5.189	+	9.835	+	9.477	+
E-4474	6.675	+	5.696	+	3.666	+	5.164	+
E-4501	4.482	+	5.557	+	6.532	+	8.911	+
E-4534	6.646	+	5.690	+	4.779	+	6.532	+
E-4535	3.968	+	4.026	+	1.812	+	4.165	+
E-4767	3.762	+	4.552	+	4.063	+	6.507	+
E-5065	4.155	+	4.417	+	3.292	+	6.326	+
Sparfloxacin	5.597	+	6.220	+	11.507	+	12.839	−
Fleroxacin	6.536	+	2.914	+	5.178	+	7.890	+
Garenoxacin	3.172	+	4.498	+	4.661	+	4.229	+
Gatifloxacin	2.807	+	4.236	+	5.433	+	8.125	+
Gemifloxacin	0.941	−	2.268	+	−1.575	−	0.198	+
Levofloxacin	0.778	−	0.176	+	2.779	+	5.358	+
Lomefloxacin	3.536	+	1.699	+	6.580	+	8.713	+
Norfloxacin	0.835	−	0.040	+	1.270	+	3.048	+
Olamufloxacin	6.884	+	8.159	+	7.029	+	7.670	+
Pazufloxacin	3.439	+	3.256	+	3.095	+	5.559	+
Pefloxacin	0.434	−	0.003	+	0.251	+	3.216	+
PGE-4175997	−0.023	−	2.379	+	3.933	+	2.834	+
PGE-9509924	2.682	+	2.674	+	−1.660	−	1.498	+
Sitafloxacin	8.074	+	8.583	+	9.195	+	9.874	+
Temafloxacin	2.727	+	2.363	+	9.035	+	7.899	+
Tosufloxacin	4.105	+	2.931	+	1.552	+	−0.242	−
Ulifloxacin	2.050	+	−1.955	−	2.920	+	8.097	+
WIN57273	−0.958	−	4.073	+	1.545	+	1.421	+

^1^ Value of the DF for each compound. ^2^ The compounds are classified as active (+) if its DF value is within its highest expectancy range or inactive (−) for values out of this range.

**Table 6 biomolecules-10-01343-t006:** Results obtained after combining LDA and PDD for DF1-4. Training group: inactives.

Compound	DF1	Clas_DF1_	DF2	Clas_DF2_	DF3	Clas_DF3_	DF4	Clas_DF4_
Cinchophen	0.043	−	−1.933	−	−6.168	−	−6.935	−
Ferron	−3.755	−	−7.099	−	−7.320	−	−6.572	−
Inact2	−4.096	−	3.663	+	1.145	+	1.959	+
Inact3	−4.735	−	−2.664	−	−8.127	−	−6.905	−
Inact6	−8.868	−	−5.489	−	−9.444	−	−8.121	−
Inact7	−7.901	−	−5.529	−	−10.331	−	−6.187	−
Inact8	−1.658	−	−0.025	−	−7.277	−	−2.580	−
Inact9	1.536	−	−0.030	−	−2.731	−	3.266	+
Inact10	−1.076	−	−1.284	−	−7.940	−	−2.100	−
Inact11	−1.201	−	−0.817	−	1.713	+	2.628	+
Inact12	−0.806	−	2.448	+	−0.090	−	2.777	+
Inact14	−1.317	−	−3.994	−	−8.495	−	−6.188	−
Inact15	−1.072	−	−3.560	−	−8.141	−	−6.040	−
Inact17	−2.475	−	−3.917	−	−8.041	−	−6.189	−
Inact18	−2.222	−	−3.484	−	−7.687	−	−6.032	−
Inact19	−2.319	−	−0.060	−	−7.177	−	−5.624	−
Inact21	−2.536	−	−3.437	−	−7.553	−	−5.967	−
Inact22	−3.930	−	−8.739	−	−10.208	−	−8.742	−
Inact23	−5.856	−	−8.647	−	−9.672	−	−8.708	−
Inact25	−4.954	−	−6.981	−	−8.297	−	−10.772	−
Inact26	−3.143	−	−5.409	−	−9.426	−	−6.349	−
Inact27	−4.655	−	−6.235	−	−7.203	−	−9.497	−
Inact28	−8.319	−	−7.957	−	−0.419	−	−3.301	−
Inact29	−8.432	−	−8.115	−	−10.453	−	−12.242	−
Inact30	−5.561	−	−6.046	−	−10.914	−	−12.054	−
Inact32	−4.364	−	−5.992	−	−8.236	−	−7.927	−
Inact33	−2.073	−	−5.849	−	−8.780	−	−8.566	−
Inact34	−6.510	−	−5.899	−	−7.985	−	−10.005	−
Inact35	−5.313	−	−4.601	−	−3.576	−	−4.415	−
Inact36	−3.528	−	−4.374	−	−3.124	−	−4.794	−
Inact38	−6.565	−	−4.384	−	−5.731	−	−7.688	−
Inact40	−2.240	−	−4.083	−	−5.346	−	−6.583	−
Inact41	0.090	−	−1.918	−	−9.637	−	−6.723	−
Inact42	0.581	−	0.045	+	−6.066	−	−8.086	−
Naptalam	−2.753	−	−2.341	−	−4.794	−	−4.237	−
PGE-5215205	−2.817	−	−1.264	−	−6.422	−	−4.291	−
PGE-6116542	0.086	−	−2.572	−	−6.091	−	−3.275	−
Quinoline	−10.175	−	−2.702	−	−8.621	−	−3.372	−

**Table 7 biomolecules-10-01343-t007:** Results obtained after combining LDA and PDD for DF1-4. Test group.

Actives								
Compound	DF1	Clas_DF1_	DF2	Clas_DF2_	DF3	Clas_DF3_	DF4	Clas_DF4_
A-80556	4.249	+	2.493	+	3.802	+	2.921	+
Balofloxacin	2.040	+	4.199	+	4.580	+	7.367	+
BMS-433366	0.976	−	2.655	+	1.483	+	4.431	+
Ciprofloxacin	1.862	−	3.858	+	2.035	+	4.915	+
DK-507k	9.830	+	8.442	+	7.892	+	8.832	+
DV-7751a	5.352	+	0.246	+	5.891	+	7.227	+
Enoxacin	0.457	−	0.472	+	1.589	+	2.373	+
Grepafloxacin	4.281	+	4.561	+	6.196	+	8.839	+
Moxifloxacin	3.323	+	4.106	+	3.061	+	6.959	+
PGE-9262932	2.160	+	2.632	+	−3.541	−	0.423	+
Rufloxacin	−3.376	−	−0.296	−	1.319	+	5.263	+
Trovafloxacin	6.145	+	5.586	+	4.622	+	4.501	+
**Inactives**								
**Compound**	**DF1**	**Clas_DF1_**	**DF2**	**Clas_DF2_**	**DF3**	**Clas_DF3_**	**DF4**	**Clas_DF4_**
Inact1	−6.386	−	−1.796	−	−7.553	−	−9.707	−
Benzoxiquine	−5.051	−	−3.857	−	−10.207	−	−10.066	−
Inact4	0.031	−	−1.528	−	−5.412	−	−2.152	−
Inact5	0.447	−	−1.528	−	−5.772	−	−2.118	−
Inact13	−0.877	−	−0.185	−	−7.770	−	−5.683	−
Inact16	−2.019	−	−0.107	−	−7.313	−	−5.692	−
Inact20	−2.789	−	−3.871	−	−7.907	−	−6.123	−
KB-5246	−8.966	−	−10.697	−	−16.779	−	−16.688	−
Inact31	−4.297	−	−5.964	−	−7.199	−	−8.047	−
Inact37	−2.737	−	−4.380	−	−10.664	−	−11.606	−
Inact39	−5.397	−	−4.451	−	−1.457	−	−3.554	−

**Table 8 biomolecules-10-01343-t008:** FDA approved drugs with identified antibacterial activity.

Drug	Approved Use	Ref.
Niclosamide	Anthelminthic	Imperi et al. [26]
Pentamidine	Antiprotozoal	Stokes et al. [27]
Ivacaftor	Anticystic fibrosis	Thakare et al. [28]
DPIC	Nitric oxide synthase inhibitor	Pandey et al. [29]
Disulfiram	Anti-alcoholic	Thakare et al. [30]
Ebelsen	Anti-inflammatory	Thangamani et al. [31]

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
