# Peer review of "Molecular Topology for the Discovery of New Broad-Spectrum Antibacterial Drugs"

_biomolecules, 2020, doi:10.3390/biom10091343_

Round 1

Reviewer 1 Report

The authors present an interesting approach to describe and then classify organic compounds according to their antibacterial activity. They focus on quinolones, a broad and relevant class of antibiotics, and provide a model that can be used to expand the exploration of new antibiotic molecules. The manuscript is well-written, and the problem being addressed is well-defined, as is the methodology. 

The one major issue is that I do suspect that the impact of the proposed model seems to be overstated. In particular, the claim made in the abstract that this model can reliably predict "antibacterial activity in any organic compound" is significantly overstated, given that organic compounds with novel mechanisms of action would not be detected by a model that was not trained with previous examples of that mechanism of action. This model was only trained with quinolones, which mostly target the ligase activity of topoisomerases, hindering DNA replication in bacteria.

Could this model predict the antibacterial activity of a drug that targets the bacterial ribosome, for example?

While the authors make a laudable effort to present their model, training set, parameters in a clear and objective way (for example, with the SI material and discriminant functions described in the results), it would be very beneficial for the reader to have a clear explanation of the indices used to build the four models. As of now, the reader only understands the model in the discussion section, but the methods or results section could host some explanations of the kinds of indices available in the dataset (charge, electrotopological‐state, connectivity, etc).

The authors make a compelling case for the use of molecular topology in drug discovery, however, conclude their manuscript with a single short paragraph of discussions, making the manuscript more reminiscent of a report than a scientific article. How can this be connected to the development of new drugs? Can this research guide the search for new compounds, or is this only compatible with random searches? In a single sentence, the authors indicate that the method can be helpful for the analysis of large databases, but how? Is the determination of molecular topology features fast or scalable? Clearly there is more to be unpacked after the described work.

Finally, no discussion is made connecting the models with mechanism of actions, even though the authors indicate that similar methodologies have been used to determine antibacterial activity of other classes of drugs. Could this method be used to predict a mechanism of action of new compounds, such as in bioprospection?

Minor items:

  • The achronym "QSAR" was used in line 42 but never defined.
  • In line 50, "the so‐called topological indices (TI)" are not defined. They are "so-called" by whom?
  • In line 140, the sentence "Three functions (DF1‐4)" was probably meant to be "Three functions (DF1‐3)".

Author Response

Response to Reviewer 1 Comments

Point 1: The authors present an interesting approach to describe and then classify organic compounds according to their antibacterial activity. They focus on quinolones, a broad and relevant class of antibiotics, and provide a model that can be used to expand the exploration of new antibiotic molecules. The manuscript is well-written, and the problem being addressed is well-defined, as is the methodology.

Response 1: We want to thank the reviewer’s comments, which we believe are fair and will help us improve our work. In the next lines we comment the changes made based on your suggestions. We would like you to known we remain at your disposal to solve any doubt or take into consideration any appreciation about our work.

Point 2: The one major issue is that I do suspect that the impact of the proposed model seems to be overstated. In particular, the claim made in the abstract that this model can reliably predict "antibacterial activity in any organic compound" is significantly overstated, given that organic compounds with novel mechanisms of action would not be detected by a model that was not trained with previous examples of that mechanism of action. This model was only trained with quinolones, which mostly target the ligase activity of topoisomerases, hindering DNA replication in bacteria. Could this model predict the antibacterial activity of a drug that targets the bacterial ribosome, for example?

Response 2: The model is directly applicable to quinolones. By using active compounds in vivo, when applied on new quinolones, you will select quinolones with similar pharmacokinetic and pharmacodynamic properties with a high percentage of success.

On the other hand, as the topological model considers the entire molecule, not only substituents on a common core, it has been proven that these models are applicable to libraries of compounds with structural diversity but that maintain adequate topological indices. Although the success rate may decrease, greater coverage of the chemical space provides a wider range (in terms of chemical diversity) from which to choose possible active compounds, which could in a further step be optimized using structure-activity relationships.

Many authors have applied this procedure to compounds with structural diversity to search for compounds with antibacterial activity, just to name a few:

Pereira F, Latino DA, Gaudencio SP. A chemoinformatics approach to the discovery of lead-like molecules from marine and microbial sources en route to antitumor and antibiotic drugs. Mar Drugs. 2014;12:757–778.

Bueso-Bordils JI, Perez-Gracia MT, Suay-Garcia B, et al. Topological pattern for the search of new active drugs against methicillin resistant Staphylococcus aureus. Eur J Med Chem. 2017;138:807–815.

Speck-Planche A, Cordeiro MN. Computer-aided discovery in antimicrobial research: In silico model for virtual screening of potent and safe anti-pseudomonas agents. Comb Chem High Throughput Screen. 2015;18:305–314.

He Y, He X. Molecular design and genetic optimization of antimicrobial peptides containing unnatural amino acids against antibioticresistant bacterial infections. Biopolymers. 2016;106:746–756.

Pereira F, Latino DA, Gaudencio SP. QSAR-assisted virtual screening of lead-like molecules from marine and microbial natural sources for antitumor and antibiotic drug discovery. Molecules. 2015;20:4848–4873.

Hodyna D, Kovalishyn V, Rogalsky S, Blagodatnyi V, Petko K, Metelytsia L. Antibacterial activity of imidazolium-based ionic liquids investigated by QSAR modeling and experimental studies. Chem Biol Drug Des. 2016;88:422–433.

Based on your comment, we have decided to include the following paragraph in the “1. Introduction” section (line 60):

The discriminant equations obtained can be applied both in new quinolones not used in the study as in molecules that have no structural relationship, since it will select those that have a similar mathematical-topological relationship.

Point 3: While the authors make a laudable effort to present their model, training set, parameters in a clear and objective way (for example, with the SI material and discriminant functions described in the results), it would be very beneficial for the reader to have a clear explanation of the indices used to build the four models. As of now, the reader only understands the model in the discussion section, but the methods or results section could host some explanations of the kinds of indices available in the dataset (charge, electrotopological‐state, connectivity, etc).

Response 3: Due to the methodology, the way of entering the indices in terms of number and type gives rise to different equations with different statistical parameters. The number of variations is very high, so explaining each of them would make the work excessively extensive. In this type of work, only those equations that are selected because they have adequate statistical parameters are explained. We respectfully feel that further explanation of each type of topological index is beyond the scope of our research article.

Point 4: The authors make a compelling case for the use of molecular topology in drug discovery, however, conclude their manuscript with a single short paragraph of discussions, making the manuscript more reminiscent of a report than a scientific article. How can this be connected to the development of new drugs? Can this research guide the search for new compounds, or is this only compatible with random searches? In a single sentence, the authors indicate that the method can be helpful for the analysis of large databases, but how? Is the determination of molecular topology features fast or scalable? Clearly there is more to be unpacked after the described work.

Response 4: We deeply appreciate the comment of the reviewer. We have decided to modify the paragraph in the “5. Conclusions” section (line 296-317), which now remains as follows:

Currently, the development of antibiotic resistance in microorganisms is one of the most important problems that have appeared in recent years in the treatment of infectious diseases. Molecular topology has proven to be a useful methodology for identifying new compounds with antibacterial activity. By combining it with LDA, four statistically significant discriminant equations with a very good classification success rate have been obtained. The stability of the selected functions is supported by external and internal validation studies, as well as by all the statistical parameters that have been used in their selection, which can be classified as very satisfactory in all cases. These equations, used both as predictive models or as filters, can be extremely useful to find new molecules with antibacterial activity and even for use in drug repositioning, where we find thousands of candidate compounds to be tested for new pharmacological activities.

We can conclude that the discriminant functions obtained confirm molecular topology as a powerful, cost-effective and useful tool in the prediction of antibacterial activity. These equations, used either as predictive models or as filters such as Lipinski’s, can be extremely useful to find new molecules with antibacterial activity and even for use in drug repositioning, where we find thousands of candidate compounds to be tested for new pharmacological activities. With this information, it is simpler to understand how certain non-obvious molecular properties described by topological indices may affect the antibacterial activity of a compound. One additional advantage of molecular topology is that it allows virtual screening of large databases in a short time for the search of possible antibacterial compounds. Moreover, this methodology also allows the prediction of pharmacokinetic and toxicological properties in order to identify safer and more efficient drugs.

Point 5: Finally, no discussion is made connecting the models with mechanism of actions, even though the authors indicate that similar methodologies have been used to determine antibacterial activity of other classes of drugs. Could this method be used to predict a mechanism of action of new compounds, such as in bioprospection?

Response 5: Virtual screening is divided into two types: receptor-based and ligand-based screening. In the first type, the affinity and geometry of ligand-receptor interactions are predicted. These techniques are frequently used for the virtual screening of large libraries of compounds and, by classifying the results, it proposes structural hypotheses in relation to the molecular inhibition of the therapeutic target studied. However, it was observed that some molecules with bioactivity similar to that of the reference molecule, but with a different molecular skeleton, were rarely detected (scaffold hopping).

It is in these cases when the second type becomes especially useful, since it is based on the use of active compounds as a reference, focusing on the comparative analysis of other compounds with different functional groups. The fact that QSAR models can be used to predict the activity of structurally different molecules has allowed the identification of lead drugs outside of the conventional antibacterial families by searching for structural, topological, or pharmacophore similarities.

Minor items:

Point 6: The achronym "QSAR" was used in line 42 but never defined.

Response 6: QSAR stands for Quantitative Structure-Activity Relationships. We’ve included these words and wrote QSAR in brackets just afterwards (line 45).

Point 7: In line 50, "the so‐called topological indices (TI)" are not defined. They are "so-called" by whom?

Response 7: The word “so-called” has been deleted from the manuscript as suggested.

Point 8: In line 140, the sentence "Three functions (DF1‐4)" was probably meant to be "Three functions (DF1‐3)".

Response 8: The reviewer is right; we have duly corrected the manuscript based on his comment (line 197).

Reviewer 2 Report

Bueso-Bordils et al. present a QSAR study on quinolones (and related compounds), a class of compounds established as anti-bacterials. Importantly, the mode of action of quinolones is only vaguely understood, hence it is difficult to rationalize the observed structure-activity relationships.

I this work, the authors collect a set of 99 quinolones with annotated bioactivity data (i.e. anti-bacterial activity) from the literature and derive simple, linear models from that. The authors claim that these models are able to discriminate active from inactive compounds.

Frankly, I am having a very hard time trying to identify any merit of this work. Firstly, I don’t see any evidence that the models are indeed useful for any prospective work (any prospective use of the models is missing in this work, and I believe it would be crucial in order to render this work acceptable for publication in Biomolecules).

Secondly, I don’t see any evidence that the models presented in this work can (or, could) contribute in any way to a better understanding of the structure-activity-relationships of quinolones. Any discussion towards this direction is missing (which is not also because of a lack of available experimental data).

Thirdly, I see major gaps in the presented results, for example, I see major issues with the work’s reproducibility (e.g. based on which exact criteria has the literature research been conducted, and based on what criteria were compounds decided to (not) be included in the dataset?) and the depth of the analysis (e.g. how can the models be interpreted? what is the relationship between model performance and the structural similarity between the test compounds and the compounds present in the training data? in what cases to the models fail (what is the applicability domain of the models?) etc.)

Fourthly, and equally importantly, the study lacks any comparison to other modeling approaches.

In short, I clearly cannot recommend the publication of this work in any form in any peer-reviewed scientific journal.

Author Response

Response to Reviewer 2 Comments

Point 1: Bueso-Bordils et al. present a QSAR study on quinolones (and related compounds), a class of compounds established as anti-bacterials. Importantly, the mode of action of quinolones is only vaguely understood, hence it is difficult to rationalize the observed structure-activity relationships.

Response 1: We deeply appreciate the comment of the reviewer. It is complicated to focus on the mode of action of quinolones since many of the indices have a more mathematical than structural meaning, but we have tried to explain the structural meaning as much as possible in each of the equations.

We have decided to include the following sentence in the “Abstract” section (line 20):

From the four discriminant functions it can be extracted that the presence of sp3 carbons, ramifications and secondary amine groups in a molecule enhance antibacterial activity, whereas the presence of 5-member rings, sp2 carbons and sp2 oxygens hinder it.

We have also included the following paragraphs and table in the “4. Discussion” section (line 226), along with their corresponding bibliographic references:

Molecular topology transforms the structure into numbers by applying more or less complex mathematical functions [4], so it is generally difficult to draw structural conclusions from mathematical equations, but from the four discriminant functions it can be extracted that the presence of sp3 carbons, ramifications and secondary amine groups in a molecule enhance antibacterial activity, whereas the presence of 5-member rings, sp2 carbons and sp2 oxygens hinder it.

Furthermore, these DFs can be used for the development of new antibacterial drugs in different ways. Firstly, QSAR models are traditionally used to optimize the activity of already existing molecules. Along these lines, Wang et al. [25] developed a QSAR model based on cinnamaldehyde-amino acid Schiff base compounds, a family of newly discovered compounds that exhibit good broad-spectrum antibacterial activity, which was then used to design and synthesize three new compounds with comparable antibacterial activity to that of ciprofloxacin. Similarly, De Bruijn et al. [26] used the information derived from a QSAR model to develop novel lead compounds with a 1,4-benzoxazin-3-one scaffold which were reported to be up to 5 times more active than any of the analogue compounds used to build the model. In the same way that these authors used the topological information provided by their models to increase the activity of a family of compounds, the DFs built in this work also provide information which could be useful to optimize the activity of antibacterial quinolones.

The second main approach by which these DFs could aid in the discovery of new antibacterial compounds is drug repurposing. There is extensive literature in which using QSAR models to reposition approved drugs is posed as a key tool in the development of new drug-disease associations [27,28]. More specifically, this approach has been already successfully used to identify antibacterial activity in a number of drugs approved for different purposes (Table 8) [29-34].

Table 8. FDA approved drugs with identified antibacterial activity.

DRUG

APPROVED USE

REF

Niclosamide

Anthelminthic

Imperi et al. [29]

Pentamidine

Antiprotozoal

Stokes et al. [30]

Ivacaftor

Anticystic fibrosis

Thakare et al. [31]

DPIC

Nitric oxide synthase inhibitor

Pandey et al. [32]

Disulfiram

Anti-alcoholic

Thakare et al. [33]

Ebelsen

Anti-inflammatory

Thangamani et al. [34]

  1. Wang, H.; Jiang, M.; Sun, F.; Li, S.; Hse, C.Y.; Jin, C. Screening, Synthesis, and QSAR Research on Cinnamaldehyde-Amino Acid Schiff Base Compounds as Antibacterial Agents. Molecules. 2018, 23(11), 3027. doi: 10.3390/molecules23113027
  2. De Bruijn, W.J.C.; Hageman, J.A.; Araya-Cloutier, C.; Gruppen, H.; Vincken, J.P. QSAR of 1,4-benzoxazin-3-one antimicrobials and their drug design perspectives. Bioorg. Med. Chem. 2018, 26(23-24), 6105-6114. doi: 10.1016/j.bmc.2018.11.016
  3. Khalid, Z.; Sezerman, O.U. Computational drug repurposing to predict approved and novel drug-disease associations. J. Mol. Graph. Model. 2018, 85, 91-96. doi: 10.1016/j.jmgm.2018.08.005
  4. Jarada, T.M.; Rokne, J.G.M Alhajj, R. A review of computational drug repositioning: strategies, approaches, opportunities, challenges, and directions. 2020, 12, 46. doi: 10.1186/s13321-020-00450-7
  5. Imperi, F., Massai, F., Ramachandran, P.C.; Longo, F.; Zennaro, E.; Rampioni, G.; Visca, P.; Leoni, L. New life for an old drug: the anthelmintic drug niclosamide inhibits Pseudomonas aeruginosa quorum sensing. Antimcirob. Agents Chemother. 2013, 57, 996-1005. doi: 10.1128/AAC.01952-12
  6. Stokes, J.M.; MacNair, C.R.; Ilyas, B.; French, S.; Côté, J.P.; Bouwman, C.; Farha, M.A.; Sieron, A.O.; Whitfield, C.; Coombes, B.K.; Brown, E.D. Pentamidine sensitizes Gram-negative pathogens to antibiotics and overcomes acquired colistin resistance. Nat. Microbiol. 2017, 6(2), 17028. doi: 10.1038/nmicrobiol.2017.28
  7. Thakare, R.; Singh, A.K.; Das, S.; Vasudevan, N.; Jachak, G.R.; Reddy, D.S.; Dasgupta, A.; Chopra, S. Repurposing Ivacaftor for treatment of Staphylococcus aureus infections. Int. J. Antimicrob. Agents. 2017, 50(3), 389-392. doi: 10.1016/j.ijantimicag. 2017.03.020
  8. Pandey, M.; Singh, A.K.; Thakare, R.; Talwar, S.; Karaulia, P.; Dasgupta, A.; Chopra, S.; Pandey, A.K. Diphenyleneiodonium chloride (DPIC) displays broad-spectrum bactericidal activity. Sci. Rep. 2017, 7(1), 11521. doi: 10.1038/s41598-017-11575-5
  9. Thakare, R.; Shulka, M.; Kaul, G.; Dasgupta, A.; Chopra, S. Repurposing disulfiram for treatment of Staphylococcus aureus infections. Int. J. Antimicrob. Agents. 2019, 53(6), 715. doi: 10.1016/j.ijantimicag.2019.03.024
  10. Thangamani, S.; Younis, W.; Seleem, M.N. Repurposing ebselen for treatment of multidrug-resistant staphylococcal infections. Sci. Rep. 2015, 5, 11596. doi: 10.1038/srep11596

Point 2: I this work, the authors collect a set of 99 quinolones with annotated bioactivity data (i.e. anti-bacterial activity) from the literature and derive simple, linear models from that. The authors claim that these models are able to discriminate active from inactive compounds.

Frankly, I am having a very hard time trying to identify any merit of this work. Firstly, I don’t see any evidence that the models are indeed useful for any prospective work (any prospective use of the models is missing in this work, and I believe it would be crucial in order to render this work acceptable for publication in Biomolecules).

Response 2: There are two ways to use these equations:

  1. a) As a prediction model, this is, a molecule will have a higher probability of having antibacterial activity if it meets the appropriate values of DF1 and/or DF2 and/or DF3 and/or DF4. It should be noted that the greater the number of DFs, the more restrictive the model will be.
  2. b) As filters. In drug design, it is sometimes difficult to choose which molecules pass from the virtual space to the real one. On many occasions we find virtual libraries of very numerous theoretically active compounds. Therefore, only those that have a high probability of success should be the ones chosen to be synthesized and tested. As with Lipinski's rule, these equations can be extremely useful as filters to increase the probability of success.

We have also decided to modify the paragraph in the “5. Conclusions” section (lines 296-317), which now remains as follows:

Currently, the development of antibiotic resistance in microorganisms is one of the most important problems that have appeared in recent years in the treatment of infectious diseases. Molecular topology has proven to be a useful methodology for identifying new compounds with antibacterial activity. By combining it with LDA, four statistically significant discriminant equations with a very good classification success rate have been obtained. The stability of the selected functions is supported by external and internal validation studies, as well as by all the statistical parameters that have been used in their selection, which can be classified as very satisfactory in all cases. These equations, used both as predictive models or as filters, can be extremely useful to find new molecules with antibacterial activity and even for use in drug repositioning, where we find thousands of candidate compounds to be tested for new pharmacological activities.

We can conclude that the discriminant functions obtained confirm molecular topology as a powerful, cost-effective and useful tool in the prediction of antibacterial activity. These equations, used either as predictive models or as filters such as Lipinski’s, can be extremely useful to find new molecules with antibacterial activity and even for use in drug repositioning, where we find thousands of candidate compounds to be tested for new pharmacological activities. With this information, it is simpler to understand how certain non-obvious molecular properties described by topological indices may affect the antibacterial activity of a compound. One additional advantage of molecular topology is that it allows virtual screening of large databases in a short time for the search of possible antibacterial compounds. Moreover, this methodology also allows the prediction of pharmacokinetic and toxicological properties in order to identify safer and more efficient drugs.

Point 3: Secondly, I don’t see any evidence that the models presented in this work can (or, could) contribute in any way to a better understanding of the structure-activity-relationships of quinolones. Any discussion towards this direction is missing (which is not also because of a lack of available experimental data).

Response 3: The usefulness of this type of mathematical-topological equations has been explained in previous answers 1 and 2.

Point 4: Thirdly, I see major gaps in the presented results, for example, I see major issues with the work’s reproducibility (e.g. based on which exact criteria has the literature research been conducted, and based on what criteria were compounds decided to (not) be included in the dataset?) and the depth of the analysis (e.g. how can the models be interpreted? what is the relationship between model performance and the structural similarity between the test compounds and the compounds present in the training data? in what cases to the models fail (what is the applicability domain of the models?) etc.)

Response 4: The model is directly applicable to quinolones. By using active compounds in vivo, when applied on new quinolones, you will select quinolones with similar pharmacokinetic and pharmacodynamic properties with a high percentage of success.

On the other hand, as the topological model considers the entire molecule, not only substituents on a common core, it has been proven that these models are applicable to libraries of compounds with structural diversity but that maintain adequate topological indices. Although the success rate may decrease, greater coverage of the chemical space provides a wider range (in terms of chemical diversity) from which to choose possible active compounds, which could in a further step be optimized using structure-activity relationships.

Many authors have applied this procedure to compounds with structural diversity to search for compounds with antibacterial activity, just to name a few:

Pereira F, Latino DA, Gaudencio SP. A chemoinformatics approach to the discovery of lead-like molecules from marine and microbial sources en route to antitumor and antibiotic drugs. Mar Drugs. 2014;12:757–778.

Bueso-Bordils JI, Perez-Gracia MT, Suay-Garcia B, et al. Topological pattern for the search of new active drugs against methicillin resistant Staphylococcus aureus. Eur J Med Chem. 2017;138:807–815.

Speck-Planche A, Cordeiro MN. Computer-aided discovery in antimicrobial research: In silico model for virtual screening of potent and safe anti-pseudomonas agents. Comb Chem High Throughput Screen. 2015;18:305–314.

He Y, He X. Molecular design and genetic optimization of antimicrobial peptides containing unnatural amino acids against antibioticresistant bacterial infections. Biopolymers. 2016;106:746–756.

Pereira F, Latino DA, Gaudencio SP. QSAR-assisted virtual screening of lead-like molecules from marine and microbial natural sources for antitumor and antibiotic drug discovery. Molecules. 2015;20:4848–4873.

Hodyna D, Kovalishyn V, Rogalsky S, Blagodatnyi V, Petko K, Metelytsia L. Antibacterial activity of imidazolium-based ionic liquids investigated by QSAR modeling and experimental studies. Chem Biol Drug Des. 2016;88:422–433.

Due to the methodology, the way of entering the indices in terms of number and type gives rise to different equations with different statistical parameters. The number of variations is very high, so explaining each of them would make the work excessively extensive. In this type of work, only those equations that are selected because they have adequate statistical parameters are explained.

Based on your comment, we have decided to include the following paragraph in the “1. Introduction” section (line 60):

The discriminant equations obtained can be applied both in new quinolones not used in the study as in molecules that have no structural relationship, since it will select those that have a similar mathematical-topological relationship.

Point 5: Fourthly, and equally importantly, the study lacks any comparison to other modeling approaches.

Response 5: Virtual screening is divided into two types: receptor-based and ligand-based screening. In the first type, the affinity and geometry of ligand-receptor interactions are predicted. These techniques are frequently used for the virtual screening of large libraries of compounds and, by classifying the results, it proposes structural hypotheses in relation to the molecular inhibition of the therapeutic target studied. However, it was observed that some molecules with bioactivity similar to that of the reference molecule, but with a different molecular skeleton, were rarely detected (scaffold hopping).

It is in these cases when the second type becomes especially useful, since it is based on the use of active compounds as a reference, focusing on the comparative analysis of other compounds with different functional groups. The fact that QSAR models can be used to predict the activity of structurally different molecules has allowed the identification of lead drugs outside of the conventional antibacterial families by searching for structural, topological, or pharmacophore similarities.

Point 6: In short, I clearly cannot recommend the publication of this work in any form in any peer-reviewed scientific journal.

Response 6: We deeply respect your opinion but hope that our clarifications and improvements based on your comments might help changing your mind.

Reviewer 3 Report

Bueso-Bordils et al. provides a LDA (linear discriminant analysis) for the prediction of bio-active compounds against bacteria (antibiotics). While the work is sounded and with specific aims, it lacks a thorough presentation of the results and an appropriate discussion of them. My recommendation is to provide a more structured introduction of the results in combination with a more general discussion. To my eyes, many of the details provided in the discussion can be easily moved into the results and leave the discussion for an integration/comparison with other approaches published elsewhere.

I would like to also point to some specific concerns which may require further clarifications:

1.- The word "topology" is overall provided as a way to identify the training molecules. However, it seems like the authors are using more "chemical descriptors" rather than spatial "topological" descriptors. I got confused as in line 79 it is clearly mentioned that these descriptors do not contain 3D parameters. This should be clarified.

2.- Lines 41-48, it is not clear to me how QSAR properties can be integrated in order to design new compounds with pharmacological activity. QSAR properties provide inherent physics-chemical related features which don't carry any bio-activity at all. You first need to know that a molecule is bio-active by means of collected-measured-stored experimental data. Whitout that information I doubt you can know if a molecule is bio-active. I believe that is the reason why authors are using MIC as an important property in the first place. This has to be clarified.

3.- Given the introduction, seems like that the overall objective of the work is to get a tool to expand the therapeutic arsenal, however results only suggests that the method can discriminate between active and non-active compounds for known molecules. It is not clear if authors were able to achieve the objective or whether or not there was an attempt to generate "novel" molecules with bio-active properties. This should be discussed.

4.- It should be presented and discussed other approaches published elsewhere: pros/cons and why LDA is a good alternative. For instance:

Mansbach, R. A., Leus, I. V., Mehla, J., López, C. A., Walker, J. K., Rybenkov, V. V., et al. (2020). Machine Learning Algorithm Identifies an Antibiotic Vocabulary for Permeating Gram-Negative Bacteria. Journal of Chemical Information and Modeling, 60(6), 2838–2847. http://doi.org/10.1021/acs.jcim.0c00352

tackles the same issue, although more specific towards certain molecular properties (permeation) attaining drug resistance. 

5.- Some of the decisions for selecting the training molecules are somehow arbitrary (lines 66-75), is there a rational for such discrimination?

6.-  Limitations and drawbacks of the method should be further discussed. It is not clear if the training set is enough, or the prediction could be improved by adding more candidates. Similarly, and to properly assess the predictability, the method should be applied to a larger library with molecules sharing similar descriptors but never used as training candidates.

Author Response

Response to Reviewer 3 Comments

Point 1: Bueso-Bordils et al. provides a LDA (linear discriminant analysis) for the prediction of bio-active compounds against bacteria (antibiotics). While the work is sounded and with specific aims, it lacks a thorough presentation of the results and an appropriate discussion of them. My recommendation is to provide a more structured introduction of the results in combination with a more general discussion. To my eyes, many of the details provided in the discussion can be easily moved into the results and leave the discussion for an integration/comparison with other approaches published elsewhere.

Response 1: We deeply appreciate the comment of the reviewer. Based on his observations, we have relocated lines 251-294 (including figures 1-4 and tables 5-7) from the “4. Discussion” section to line 143 in the “3. Results” section.

We have decided to include the following sentence in the “Abstract” section (line 20):

From the four discriminant functions it can be extracted that the presence of sp3 carbons, ramifications and secondary amine groups in a molecule enhance antibacterial activity, whereas the presence of 5-member rings, sp2 carbons and sp2 oxygens hinder it.

We have also included the following paragraphs and table in the “4. Discussion” section (line 226), along with their corresponding bibliographic references:

Molecular topology transforms the structure into numbers by applying more or less complex mathematical functions [4], so it is generally difficult to draw structural conclusions from mathematical equations, but from the four discriminant functions it can be extracted that the presence of sp3 carbons, ramifications and secondary amine groups in a molecule enhance antibacterial activity, whereas the presence of 5-member rings, sp2 carbons and sp2 oxygens hinder it.

Furthermore, these DFs can be used for the development of new antibacterial drugs in different ways. Firstly, QSAR models are traditionally used to optimize the activity of already existing molecules. Along these lines, Wang et al. [25] developed a QSAR model based on cinnamaldehyde-amino acid Schiff base compounds, a family of newly discovered compounds that exhibit good broad-spectrum antibacterial activity, which was then used to design and synthesize three new compounds with comparable antibacterial activity to that of ciprofloxacin. Similarly, De Bruijn et al. [26] used the information derived from a QSAR model to develop novel lead compounds with a 1,4-benzoxazin-3-one scaffold which were reported to be up to 5 times more active than any of the analogue compounds used to build the model. In the same way that these authors used the topological information provided by their models to increase the activity of a family of compounds, the DFs built in this work also provide information which could be useful to optimize the activity of antibacterial quinolones.

The second main approach by which these DFs could aid in the discovery of new antibacterial compounds is drug repurposing. There is extensive literature in which using QSAR models to reposition approved drugs is posed as a key tool in the development of new drug-disease associations [27,28]. More specifically, this approach has been already successfully used to identify antibacterial activity in a number of drugs approved for different purposes (Table 8) [29-34].

Table 8. FDA approved drugs with identified antibacterial activity.

DRUG

APPROVED USE

REF

Niclosamide

Anthelminthic

Imperi et al. [29]

Pentamidine

Antiprotozoal

Stokes et al. [30]

Ivacaftor

Anticystic fibrosis

Thakare et al. [31]

DPIC

Nitric oxide synthase inhibitor

Pandey et al. [32]

Disulfiram

Anti-alcoholic

Thakare et al. [33]

Ebelsen

Anti-inflammatory

Thangamani et al. [34]

  1. Wang, H.; Jiang, M.; Sun, F.; Li, S.; Hse, C.Y.; Jin, C. Screening, Synthesis, and QSAR Research on Cinnamaldehyde-Amino Acid Schiff Base Compounds as Antibacterial Agents. Molecules. 2018, 23(11), 3027. doi: 10.3390/molecules23113027
  2. De Bruijn, W.J.C.; Hageman, J.A.; Araya-Cloutier, C.; Gruppen, H.; Vincken, J.P. QSAR of 1,4-benzoxazin-3-one antimicrobials and their drug design perspectives. Bioorg. Med. Chem. 2018, 26(23-24), 6105-6114. doi: 10.1016/j.bmc.2018.11.016
  3. Khalid, Z.; Sezerman, O.U. Computational drug repurposing to predict approved and novel drug-disease associations. J. Mol. Graph. Model. 2018, 85, 91-96. doi: 10.1016/j.jmgm.2018.08.005
  4. Jarada, T.M.; Rokne, J.G.M Alhajj, R. A review of computational drug repositioning: strategies, approaches, opportunities, challenges, and directions. 2020, 12, 46. doi: 10.1186/s13321-020-00450-7
  5. Imperi, F., Massai, F., Ramachandran, P.C.; Longo, F.; Zennaro, E.; Rampioni, G.; Visca, P.; Leoni, L. New life for an old drug: the anthelmintic drug niclosamide inhibits Pseudomonas aeruginosa quorum sensing. Antimcirob. Agents Chemother. 2013, 57, 996-1005. doi: 10.1128/AAC.01952-12
  6. Stokes, J.M.; MacNair, C.R.; Ilyas, B.; French, S.; Côté, J.P.; Bouwman, C.; Farha, M.A.; Sieron, A.O.; Whitfield, C.; Coombes, B.K.; Brown, E.D. Pentamidine sensitizes Gram-negative pathogens to antibiotics and overcomes acquired colistin resistance. Nat. Microbiol. 2017, 6(2), 17028. doi: 10.1038/nmicrobiol.2017.28
  7. Thakare, R.; Singh, A.K.; Das, S.; Vasudevan, N.; Jachak, G.R.; Reddy, D.S.; Dasgupta, A.; Chopra, S. Repurposing Ivacaftor for treatment of Staphylococcus aureus infections. Int. J. Antimicrob. Agents. 2017, 50(3), 389-392. doi: 10.1016/j.ijantimicag. 2017.03.020
  8. Pandey, M.; Singh, A.K.; Thakare, R.; Talwar, S.; Karaulia, P.; Dasgupta, A.; Chopra, S.; Pandey, A.K. Diphenyleneiodonium chloride (DPIC) displays broad-spectrum bactericidal activity. Sci. Rep. 2017, 7(1), 11521. doi: 10.1038/s41598-017-11575-5
  9. Thakare, R.; Shulka, M.; Kaul, G.; Dasgupta, A.; Chopra, S. Repurposing disulfiram for treatment of Staphylococcus aureus infections. Int. J. Antimicrob. Agents. 2019, 53(6), 715. doi: 10.1016/j.ijantimicag.2019.03.024
  10. Thangamani, S.; Younis, W.; Seleem, M.N. Repurposing ebselen for treatment of multidrug-resistant staphylococcal infections. Sci. Rep. 2015, 5, 11596. doi: 10.1038/srep11596

Point 2: I would like to also point to some specific concerns which may require further clarifications: 1.- The word "topology" is overall provided as a way to identify the training molecules. However, it seems like the authors are using more "chemical descriptors" rather than spatial "topological" descriptors. I got confused as in line 79 it is clearly mentioned that these descriptors do not contain 3D parameters. This should be clarified.

Response 2: Thank you again for your comment. When we talk about 3D descriptors, we mean descriptors that take into account the stereochemistry of the molecule and can distinguish between enantiomers. We have decided to modify the sentence “These descriptors do not contain 3D parameters” from line 84 to “These descriptors do not contain 3D parameters that distinguish between enantiomers”.

We have also added, for further clarification, the following sentence in line 88 along with the following bibliographic reference:

“Chemical pseudographs are two-dimensional objects and, therefore, so are their molecular descriptors, although they are capable of representing a high content of information about three-dimensional structures”:

Roy, K.; Das, R.N. A review on principles, theory and practices of 2D-QSAR. Curr. Drug Metab. 2014, 15, 346-379. doi: 10.2174/1389200215666140908102230

Point 3: 2.- Lines 41-48, it is not clear to me how QSAR properties can be integrated in order to design new compounds with pharmacological activity. QSAR properties provide inherent physics-chemical related features which don't carry any bio-activity at all. You first need to know that a molecule is bio-active by means of collected-measured-stored experimental data. Whitout that information I doubt you can know if a molecule is bio-active. I believe that is the reason why authors are using MIC as an important property in the first place. This has to be clarified.

Response 3: These studies provide useful information for a better understanding of how the molecular structure may affect the antibacterial activity of a compound. Having this information, it is simpler to optimize existing drugs, which is a cost-effective approach to obtain new treatments against resistant bacteria.

Quinolones and quinolone-like structures are selected and classified as active/inactive by means of their in vitro MIC values, dismissing intermediate MIC values in order to ensure the maximum possible structure-activity differences between both groups.

Point 4: 3.- Given the introduction, seems like that the overall objective of the work is to get a tool to expand the therapeutic arsenal, however results only suggests that the method can discriminate between active and non-active compounds for known molecules. It is not clear if authors were able to achieve the objective or whether or not there was an attempt to generate "novel" molecules with bio-active properties. This should be discussed.

Response 4: We deeply appreciate the comment of the reviewer. These issues have been partly addressed in response 3.

Based on your comment, we have decided to include the following paragraph in the “1. Introduction” section (line 60):

The discriminant equations obtained can be applied both in new quinolones not used in the study as in molecules that have no structural relationship, since it will select those that have a similar mathematical-topological relationship.

We have also modified the paragraph in the “5. Conclusions” section (line 296), which now remains as follows:

Currently, the development of antibiotic resistance in microorganisms is one of the most important problems that have appeared in recent years in the treatment of infectious diseases. Molecular topology has proven to be a useful methodology for identifying new compounds with antibacterial activity. By combining it with LDA, four statistically significant discriminant equations with a very good classification success rate have been obtained. The stability of the selected functions is supported by external and internal validation studies, as well as by all the statistical parameters that have been used in their selection, which can be classified as very satisfactory in all cases. These equations, used both as predictive models or as filters, can be extremely useful to find new molecules with antibacterial activity and even for use in drug repositioning, where we find thousands of candidate compounds to be tested for new pharmacological activities.

We can conclude that the discriminant functions obtained confirm molecular topology as a powerful, cost-effective and useful tool in the prediction of antibacterial activity. These equations, used either as predictive models or as filters such as Lipinski’s, can be extremely useful to find new molecules with antibacterial activity and even for use in drug repositioning, where we find thousands of candidate compounds to be tested for new pharmacological activities. With this information, it is simpler to understand how certain non-obvious molecular properties described by topological indices may affect the antibacterial activity of a compound. One additional advantage of molecular topology is that it allows virtual screening of large databases in a short time for the search of possible antibacterial compounds. Moreover, this methodology also allows the prediction of pharmacokinetic and toxicological properties in order to identify safer and more efficient drugs.

Point 5: 4.- It should be presented and discussed other approaches published elsewhere: pros/cons and why LDA is a good alternative. For instance:

Mansbach, R. A., Leus, I. V., Mehla, J., López, C. A., Walker, J. K., Rybenkov, V. V., et al. (2020). Machine Learning Algorithm Identifies an Antibiotic Vocabulary for Permeating Gram-Negative Bacteria. Journal of Chemical Information and Modeling, 60(6), 2838–2847. http://doi.org/10.1021/acs.jcim.0c00352

tackles the same issue, although more specific towards certain molecular properties (permeation) attaining drug resistance.

Response 5: Many authors have applied this procedure to compounds with structural diversity to search for compounds with antibacterial activity, just to name a few:

Pereira F, Latino DA, Gaudencio SP. A chemoinformatics approach to the discovery of lead-like molecules from marine and microbial sources en route to antitumor and antibiotic drugs. Mar Drugs. 2014;12:757–778.

Bueso-Bordils JI, Perez-Gracia MT, Suay-Garcia B, et al. Topological pattern for the search of new active drugs against methicillin resistant Staphylococcus aureus. Eur J Med Chem. 2017;138:807–815.

Speck-Planche A, Cordeiro MN. Computer-aided discovery in antimicrobial research: In silico model for virtual screening of potent and safe anti-pseudomonas agents. Comb Chem High Throughput Screen. 2015;18:305–314.

He Y, He X. Molecular design and genetic optimization of antimicrobial peptides containing unnatural amino acids against antibioticresistant bacterial infections. Biopolymers. 2016;106:746–756.

Pereira F, Latino DA, Gaudencio SP. QSAR-assisted virtual screening of lead-like molecules from marine and microbial natural sources for antitumor and antibiotic drug discovery. Molecules. 2015;20:4848–4873.

Hodyna D, Kovalishyn V, Rogalsky S, Blagodatnyi V, Petko K, Metelytsia L. Antibacterial activity of imidazolium-based ionic liquids investigated by QSAR modeling and experimental studies. Chem Biol Drug Des. 2016;88:422–433.

However, we respectfully feel that further explanation of other drug design approaches is beyond the scope of our research article.

Point 6: 5.- Some of the decisions for selecting the training molecules are somehow arbitrary (lines 66-75), is there a rational for such discrimination?

Response 6: These issues have already been addressed in response 3.

Point 7: 6.-  Limitations and drawbacks of the method should be further discussed. It is not clear if the training set is enough, or the prediction could be improved by adding more candidates. Similarly, and to properly assess the predictability, the method should be applied to a larger library with molecules sharing similar descriptors but never used as training candidates.

Response 7: These issues have already been addressed in response 1.

Round 2

Reviewer 2 Report

The authors have adequately addressed all comments.

Reviewer 3 Report

I appreciate authors for addressing my concerns.